# DOMAIN-AGNOSTIC SELF-TRAINING FOR SEMI-SUPERVISED LEARNING

## ABSTRACT

Self-training is a popular class of semi-supervised learning (SSL) methods which can be viewed as iteratively assigning pseudo labels to unlabeled data for model training. Despite its recent successes, most self-training approaches are domain-specific, relying on the predefined data augmentation schemes in a particular domain to generate reliable pseudo labels. In this paper, we propose a domain-agnostic self-training framework named DAST, which is applicable to domains where prior knowledge is not readily available. DAST consists of a contrastive learning module along with a novel two-way pseudo label generation strategy. Without the reliance of data augmentation, DAST performs supervised contrastive learning with the pseudo labels generated from the classifier to learn aligned data representations and produces the reliable pseudo labels for self-training based on the learned representations. From an expectation maximization (EM) algorithm perspective, we theoretically prove that representation learning and self-training in DAST are mutually beneficial. Extensive experiments in various domains (tabular data, graphs, and images.) verify that DAST not only significantly outperforms other domain-agnostic self-training methods, but can also combine with effective domain knowledge to further boost the performance.

## 1 INTRODUCTION

The remarkable success of deep neural networks is partially attributed to the collection of large labeled datasets (Zhu, 2005; 2007). In real-world applications, however, it is extremely expensive and time-consuming for manually labeling sufficient data with high-quality annotations. To reduce the requirement for labeled data during model training, a powerful approach is semi-supervised learning (SSL) which improves the data efficiency of deep models by leveraging a large number of unlabeled samples (Lee et al., 2013; Rasmus et al., 2015). Among them, self-training has emerged as one of the most promising approaches to deal with unlabeled samples and achieved competitive performance in many domains such as image (Lee et al., 2013; Sohn et al., 2020; Chen et al., 2022), text (Yuan et al., 2016; Xie et al., 2020), and graph data (Hao et al., 2020; Wang & Leskovec, 2021). Typically, self-training approaches can be viewed as iteratively assigning pseudo labels to unlabeled samples and then feeding them as input for model training.

Despite its popularity, a challenging task of self-training is to generate the reliable pseudo labels for unlabeled samples. The simplest solution is to use the model's predictions as true labels to train against (Rosenberg et al., 2005; Lee et al., 2013), which is well-believed to be unreliable and often results in training instability (Xie et al., 2020; Chen et al., 2022). To produce high-quality pseudo labels for unlabeled samples, one line of researches employ data augmentations to generate distorted versions of unlabeled samples, and guess the labels with augmented data which constrains the generated pseudo labels to be invariant of the noise (Berthelot et al., 2019; Sohn et al., 2020; Li et al., 2021; Pham et al., 2021; Chen et al., 2022; Oh et al., 2022). Another line of works utilize contrastive learning to produce aligned data representations and generate pseudo labels under the assumption that data points closer in the embedding space are more likely to share the same ground-truth label (Chen et al., 2020b; Assran et al., 2021; Li et al., 2021; Wang et al., 2022).

Both lines of existing methods heavily rely on effective data augmentation schemes in a particular domain, like flipping and cropping in images, and expect the augmented samples would not change the data semantics. Otherwise, training a model with false positive augmented samples would generate

unreliable pseudo labels (Yue et al., 2022), which eventually causes the training error accumulation and performance fluctuations. This fact seriously prohibits existing self-training methods from broader applicability, considering the unexplored domains (e.g., tabular data) where effective augmentation are not readily available (Somepalli et al., 2021; Bahri et al., 2022) and domains (e.g., graph data) without universal data augmentations that are consistently effective over different datasets (You et al., 2021; Lee et al., 2022). Nevertheless, to the best of our knowledge, few efforts have been made to resolve the above mentioned issue.

This paper aims to generate reliable pseudo labels without the reliance on data augmentations for improving the self-training performance in unexplored domains. We design a framework named **D**omain-**A**gnostic **S**elf-**T**raining (**DAST**) with a novel two-way pseudo label generation strategy that integrates two highly dependent problems—self-training and representation learning—in a cohesive and mutually beneficial manner. First, DAST encapsulates an additional contrastive learning module which is trained along with the classification head (Section 3.2). To avoid the reliance on data augmentation, we generate the pseudo labels from the classification head to construct contrastive pairs and perform supervised contrastive learning (Khosla et al., 2020) to learn data representations. Second, we produce high-quality pseudo labels for self-training based on the learned representations from the contrastive learning module (Section 3.3). The key idea is to construct a data similarity graph by exploiting the learned representations and infer the pseudo labels using graph transductive learning. Therefore, DAST prevents each of the classification module and the representation learning module from generating and utilizing the pseudo labels on its own, which avoids the direct error accumulation and thereby improving the model performance. By interleaving the above two steps, DAST converges to a solution with a highly distinguishable representation for accurate pseudo label generation. Furthermore, based on the proposed pseudo label generation strategy, we theoretically show that the contrastive representation learning and self-training in DAST are mutually beneficial (Section 4) from an Expectation Maximization (EM) perspective.

To summarize, this paper makes the following contributions. **First**, we propose a novel domain-agnostic self-training framework DAST that eliminates the reliance of data augmentation for unexplored domains. We introduce a new two-way pseudo label generation strategy that effectively integrates the representation learning and self-training for better model performance. **Second**, we provide the intuitive explanation and theoretical analysis to show that representation learning and self-training in DAST are mutually beneficial. **Third**, we conduct extensive experiments in three domains, including tabular data, graphs and images. The results show that (i) DAST achieves average performance improvements of 1.12%, 0.62%, and 0.42% against the best domain-agnostic self-training baseline in three domains, respectively. (ii) DAST can seamlessly incorporate existing data augmentations to further improve the performance by up to 2.62% and 1% compared with the state-of-the-art baselines in graph and image datasets.

## 2 RELATED WORK

Semi-supervised learning is a mature field with a huge diversity of approaches. In this review, we only focus on the SSL methods which are closely related to DAST. Broader introductions could be found in (Chapelle et al., 2009; Zhu & Goldberg, 2009; Van Engelen & Hoos, 2020).

**Self-training for semi-supervised learning.** Self-training is a popular class of SSL approaches which can date back for decades (Scudder, 1965; McLachlan, 1975) and still keeps vibrant in recent years (Sohn et al., 2020; Pham et al., 2021; Chen et al., 2022). The core idea behind it is iteratively assigning pseudo labels to unlabeled samples and utilize them for model training. Pseudo Label (Lee et al., 2013), one representative self-training method, directly uses model predictions as pseudo labels and feeds them for model training. Mean-Teacher (Tarvainen & Valpola, 2017) designs teacher-student network and generates pseudo labels from an exponential moving average of the model. However, this paradigm suffers from the unreliability of pseudo labels since the deep models would fit to inaccurate pseudo labels, resulting in performance degeneration (Chen et al., 2022).

Recent works tackle this issue by utilizing domain-specific knowledge to generate high-quality pseudo labels. In the following, we briefly review recent self-training approaches in different domains and discuss their applicability. In computer vision, the state-of-the-art self-training approaches (Berthelot et al., 2020; Sohn et al., 2020; Oh et al., 2022) generate pseudo labels using model predictions on weakly-augmented samples (e.g., image flip) and utilize these pseudo labels as annotations

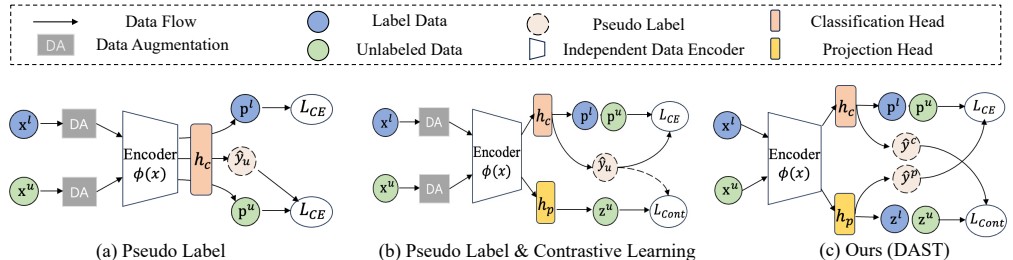

Figure 1: Comparisons on how different self-training methods generate and utilize pseudo labels. (a) Pseudo Label methods (e.g., FixMatch) rely on data augmentation to generate pseudo labels and utilize them for training the whole model. (b) Pseudo Label combined with contrastive learning (e.g., CoMatch) relies on data augmentations to generate pseudo labels and construct augmented samples for contrastive learning. The pseudo labels generated from the classification head are used for training the whole model. (c) DAST eliminates the reliance of data augmentations and decouples the generation and utilization of pseudo labels.

for the strongly-augmented version of the same image, e.g., Cutout (DeVries & Taylor, 2017), RandAugment (Cubuk et al., 2020). However, these methods operate under the assumption that flipping or coloring an image would not change its semantics, and hence they cannot be applied to domains that contain structural information or heterogeneous features (Ucar et al., 2021; Lee et al., 2022), such as graph or tabular data. For graph data, a common practice to generate pseudo labels is to utilize the local and global structural contexts in a sample (Sun et al., 2019; Hao et al., 2020). These methods cannot be applied to the domains where such structural information does not exist, such as tabular domain. Due to the aforementioned reasons, how to design a reliable self-training framework that can be applied to different domains without domain-specific knowledge remains an important and unexplored problem.

**Contrastive learning for semi-supervised learning.** Contrastive learning (Chen et al., 2020a; He et al., 2020; Wang et al., 2021) is an appealing self-supervised learning method that can exploit supervision signals from unlabeled data for representation learning. Recently, some researches (Devlin et al., 2019; Chen et al., 2020b; Li et al., 2021) have utilized contrastive learning to produce distinguishable data representations to improve the performance of semi-supervised learning. These methods still require well-defined data augmentations to create reliable positive and negative contrastive pairs, and lack theoretical guarantees for the effectiveness of utilizing contrastive learning. Different from the above methods, DAST eliminates the reliance on the data augmentations by employing the classifier's outputs to construct contrastive data pairs. We have also theoretically proved that the contrastive learning module and self-training in DAST are mutually beneficial.

## 3 METHODOLOGY

In this work, we develop an effective domain-agnostic self-training framework DAST that is applicable to arbitrary domains without the requirement of specific domain knowledge. We first describe the problem setup (Section 3.1), and then present the overview of DAST (Section 3.2) and the two-way pseudo label generation strategy (Section 3.3). The pseudo-code of DAST is shown in Algorithm 1 in Appendix B.

### 3.1 PROBLEM SETUP

For a $K$-class classification problem, we have a labeled dataset $D_L = \{(x_l, y_l) \in \mathcal{X} \times \mathcal{Y} \mid 1 \leq l \leq M\}$ of $M$ labeled samples and an unlabeled dataset $D_U = \{x_u \in \mathcal{X} \mid 1 \leq u \leq N\}$ of $N$ unlabeled samples, where $\mathcal{X}$ are observations sampled from data distribution $p(x, y)$ and $\mathcal{Y} \in \{1, 2, \ldots, K\}$ is a discrete label set. We consider the setting that the size of labeled dataset is usually much smaller than that of the unlabeled dataset, i.e., $M \ll N$. Our goal is to train an effective classifier $f_\theta : \mathcal{X} \to \mathcal{Y}$ by utilizing the labeled and unlabeled datasets without any domain knowledge.

## 3.2 OVERVIEW OF DAST

For domains where mature data augmentation is not available, the augmented samples may easily change the semantics of the original data, resulting in the uncertainty of the generated pseudo labels and contrastive pairs. As shown in Figure 1(a) and Figure 1(b), the pseudo labels in existing methods are only generated from the classification head. Once the incorrect pseudo labels and contrastive pairs are generated and used for model training, the prediction errors are accumulated in the iterative self-training process and results in the degeneration of the model's performance. To solve this issue, DAST decouples the generation and utilization of pseudo labels by introducing an additional projection head that collaborates with the classification head subtly.

**Data flow.** To begin with, we first describe the data flow in DAST. As shown in Figure 1(c), DAST introduces an additional supervised contrastive learning module by encapsulating the projection head $h_p$. The projection head $h_p$ is connected to the feature encoder $\phi(x)$ and independent with the classification head $h_c$. Given an unlabeled example $x$, we generate two pseudo labels $\hat{y}^c$ and $\hat{y}^p$ based on the label distribution $p(y|x) = h_c \circ \phi(x)$ from the classification head and data representation $z = h_p \circ \phi(x)$ from the projection head, respectively. (The details of pseudo labels generation are introduced in Section 3.3). To construct contrastive data pairs for the projection head $h_p$, we utilize $\{\hat{y}^c\}$ to select pseudo positive and negative sample pairs and train the projection head via supervised contrastive loss. Formally, given an unlabeled example $x$, the per-sample contrastive loss is:

$$\mathcal{L}_{cont}(\phi, h_p; x, \tau) = \frac{-1}{|E(x)|} \sum_{e \in E(x)} \log \frac{\exp(z^\top z_e / \tau)}{\sum_{a \in A(x)} \exp(z^\top z_a / \tau)}. \tag{1}$$

Here, $E(x) \equiv \{x_e \in D_U | \hat{y}_e^c = \hat{y}^c\}$ is the set of positive samples whose predicted labels are identical to that of $x$ according to the classification head. We denote $A(x) = D_U \backslash E(x)$ as the negative sample set and $\tau \geq 0$ is the temperature. In turn, to select the pseudo labels for training the classification head, DAST introduces a label propagation method which produces pseudo labels for unlabeled data based on the learned representations from the projection head. Let $\hat{y}^p$ denote the pseudo label of an unlabeled sample $x$. The classification head is optimized by the following loss function $\mathcal{L}_{cls}$.

$$\mathcal{L}_{cls}(\phi, h_c; x) = H(\hat{y}^p, h_c \circ \phi(x)), \tag{2}$$

where $H(y, p)$ denotes the cross-entropy between two distributions $y$ and $p$. For labeled samples $(x, y)$, we directly use its ground-truth label to construct contrastive pairs in Eq. (1) and replace $\hat{y}^p$ in Eq. (2) for model training.

**Training objective.** Putting them all together, we jointly train the data encoder $\phi$, the classification head $h_c$ as well as the projection head $h_p$. The overall loss function is:

$$\mathcal{L} = \sum_{x \in D_L} (\mathcal{L}_{cls}(\phi, h_c; x) + \mathcal{L}_{cont}(\phi, h_p; x, \tau)) + \lambda \sum_{x \in D_U} (\mathcal{L}_{cls}(\phi, h_c; x) + \mathcal{L}_{cont}(\phi, h_p; x, \tau)), \tag{3}$$

where $\lambda$ is used as the trade-off between the loss on labeled data and that on unlabeled data. Note that the projection head $h_p$ and the classification head $h_c$ are trained based on the pseudo labels produced by each other. In this way, inaccurate pseudo labels will not accumulate the prediction error in each head directly during the iterative training process.

## 3.3 TWO-WAY PSEUDO LABEL GENERATION

Thus far, we present the overall training pipeline of DAST. In this section, we propose our two-way pseudo label generation strategy. We first elaborate the details on how to select accurate pseudo labels $\{\hat{y}^c\}$ from the classification head to perform supervised contrastive learning. Then we introduce the procedure of pseudo label generation for self-training. Specifically, we construct the neighbor graph of all training samples using the learned data representations and perform the graph-based label propagation to generate high-quality pseudo labels $\{\hat{y}^p\}$ for the classification head.

**Reliable contrastive pairs for the projection head.** We utilize the pseudo labels $\hat{y}^c$ generated by the classification head to construct contrastive data pairs for the projection head. To alleviate the negative effect of false positive pairs selected by inaccurate $\hat{y}^c$, we follow the existing SSL approaches (Sohn et al., 2020; Pham et al., 2021) and adopt a confidence threshold to filter out unreliable pseudo labels,

$$\hat{y}^c = \begin{cases} \arg\max_{j \in \mathcal{Y}} p(y_j|x), & \text{if } p(y_j|x) \geq \alpha, \\ -1, & \text{otherwise}, \end{cases} \tag{4}$$

where hyperparameter $\alpha$ specifies the threshold above which a pseudo label is retained and $-1$ indicates that the pseudo label will be ignored during training. Hence, we only select confident pseudo labels to construct reliable positive and negative pairs for contrastive learning in the projection head, leading to a better clustering effect in the embedding space.

**Reliable pseudo labels for the classification head.** Given data representations obtained by enforcing contrastive learning, we now describe how to generate reliable pseudo labels for training the classification head. According to the manifold assumption that similar examples should get the same prediction (Zhu & Goldberg, 2009), we propose to produce pseudo labels $\{\hat{y}^p\}$ by assigning each sample to one label-specific cluster based on the similarity of data representations from the projection head. To achieve this, we first construct a neighbor graph using the data representations. Specifically, we map each sample $x$ in labeled $D_L$ and the unlabeled dataset $D_U$ to the representation space $z = \phi(x)$ using the feature encoder and construct the affinity matrix $G$ for a neighbor graph. Each entry $g_{ij}$ in $G$ is computed as:

$$g_{ij} = \begin{cases} \text{sim}(z_i, z_j), & \text{if } i \neq j \text{ and } z_j \in \text{NN}_k(i), \\ 0, & \text{otherwise,} \end{cases} \tag{5}$$

where $\text{NN}_k(i)$ contains $k$ nearest neighbors of sample $z_i$ and $\text{sim}(\cdot, \cdot)$ is a non-negative similarity measure, e.g., $\text{Relu}(z_i^\top z_j)$. Note that constructing the neighbor graph is efficient even for large datasets (Iscen et al., 2017). Inspired by graph-based SSL approaches (Zhou et al., 2003; Wang & Leskovec, 2021), given the neighbor graph and observed labeled data, we introduce the label propagation method to obtain pseudo labels for the remaining unlabeled samples.

Let $W := G + G^\top$ to be a symmetric non-negative adjacency matrix with zero diagonal where elements $w_{ij}$ represents the non-negative pairwise similarities between $z_i$ and $z_j$. Its symmetrically normalized counterpart is given by $\mathcal{W} = D^{-1/2}WD^{-1/2}$ where $D := diag(W1_n)$ is degree matrix and $1_n$ is all-ones vector. Here we set the observed label matrix $\hat{Y} \in R^{(N+M)*K}$ as

$$\hat{Y}_{ij} = \begin{cases} 1, & \text{if } i \in D_L \text{ and } y_i = j, \\ 1, & \text{if } i \in D_U \text{ and } \hat{y}_i^c = j, \\ 0, & \text{otherwise.} \end{cases} \tag{6}$$

Note that the rows of $\hat{Y}$ corresponding to the samples with known labels or reliable pseudo labels (from the classification head) are one-hot encoded labels. Finally, we obtain the pseudo labels by computing the diffusion matrix as $C = (I - \mu\mathcal{W})^{-1}\hat{Y}$, where $\mu \in [0, 1)$ is a hyperparameter. Similar to the previous researches (Chandra & Kokkinos, 2016), we use conjugate gradient (CG) method to solve this equation to obtain $C$. Then the pseudo label $\hat{y}_i^p$ is generated by

$$\hat{y}_{ij}^p = \begin{cases} 1, & \text{if } \beta c_{ij} + (1-\beta)\hat{Y}_{ij} \geq \alpha, \\ 0, & \text{otherwise.} \end{cases} \tag{7}$$

Here $c_{ij}$ is the $(i, j)$ element of diffusion matrix $C$ and $\beta \in (0, 1)$ is a positive constant. Note that we use both the ground-truth labels and pseudo labels $\hat{y}^c$ generated by the classification head to construct label matrix $\hat{Y}$ and infer pseudo labels $\hat{y}^p$ by moving-average between $c_{ij}$ and $\hat{Y}_{ij}$. The intuition is that (i) the contrastive embeddings are less distinguishable at the beginning and high-confidence pseudo labels generated by the classification head at early stage is indicative of the ground-truth class labels. (ii) At each step, only when pseudo labels generated by the classification head and projection head are consistent with high confidence, will they be used to update the model parameters. We conduct the experiments to empirically show that the proposed label propagation strategy could produce high-quality pseudo labels in Appendix E.

**Pseudo label updating.** The most canonical way to update $\{\hat{y}^p\}$ is to compute the neighbor graph and do the label propagation in every iteration of training. However, it would extract a computational toll and in turn hurt training latency. As a result, we update the neighbor graph at every $T$ iterations to make the trade-off between time consumption and accuracy. More discussions on the setting of $T$ are introduced in Appendix E.

## 4 THEORETICAL ANALYSIS AND EXTENSION

**Why contrastive representation learning improves self-training in DAST?** In this section, we provide the intuitive explanation and theoretical justification on why contrastive learning is useful for

generating reliable pseudo labels for self-training. Intuitively, as the contrastive loss poses a clustering effect in the embedding space, our label propagation method can further generate more precise pseudo labels $\{\hat{y}^p\}$. Likewise, after training with more precise pseudo labels $\{\hat{y}^p\}$, the classification head is more likely to produce reliable pseudo labels $\{\hat{y}^c\}$ for constructing contrastive pairs, which is a crucial part of contrastive learning. The training process converges when both components perform well.

Theoretically, inspired by the *alignment property* in contrastive learning (Wang & Isola, 2020) which intrinsically minimizes the intraclass covariance in the embedding space, we can cast DAST as an expectation-maximization algorithm that maximizes the similarity of data embeddings in the same class, which coincides with the classical clustering algorithms. Specifically, at the E-step, DAST assigns the posterior distribution of the labels to unlabeled data using the pseudo labels generated from the classification head. At the M-step, the contrastive loss maximizes the expected value of likelihood function in embedding space to concentrate the embeddings to their cluster mean direction, which is achieved by minimizing Eq. (1). Finally, each training data will be mapped to a specific cluster, which benefits both pseudo label generation process and representation learning.

**E-Step.** We assume that all training samples are independent of each other. Given the training data $x_i$, we introduce $Q_i(y_i)$ as the distribution of its label $y_i$ which satisfies $\sum_{j=1}^K Q_i(y_{ij}) = 1$ and $Q_i(y_{ij}) \geq 0$. Let $\theta$ be the model parameters (here we only consider the embeddings, i.e., $\theta = \{\phi, h_p\}$). Our goal is to maximize the likelihood below:

$$\max_{\theta} \sum_{i=1}^n \log p(x_i; \theta) = \max_{\theta} \sum_{i=1}^n \log \sum_{j=1}^K p(x_i, y_{ij}; \theta) \geq \max_{\theta} \sum_{i=1}^n \sum_{j=1}^K Q_i(y_{ij}) \log \frac{p(x_i, y_{ij}; \theta)}{Q_i(y_{ij})}. \quad (8)$$

The last step of the derivation uses Jensen's inequality. Since the $\log(\cdot)$ function is concave, the equality holds when $\frac{p(x_i, y_{ij}; \theta)}{Q_i(y_{ij})} = c$ is a constant. We have $\sum_{j=1}^K p(x_i, y_{ij}; \theta) = c * \sum_{j=1}^K Q_i(y_{ij}) = c$ and hence

$$Q_i(y_{ij}) = \frac{p(x_i, y_{ij}; \theta)}{\sum_{j=1}^K p(x_i, y_{ij}; \theta)} = p(y_{ij}|x_i, \theta), \quad (9)$$

which is the posterior class probability. To estimate $p(y_{ij}|x_i, \theta)$, we directly use the ground-truth labels in the labeled dataset as well as the high-confidence pseudo labels $\hat{y}^c$ generated by the classification head. We take one-hot prediction for $p(y_{ij}|x_i, \theta)$ which means each data inherently belongs to exactly one label and $y_{ij} = 0$ if $j \neq \hat{y}_i^c$.

**M-Step.** At this step, we aim at maximizing the likelihood Eq. (8) under the assumption that the posterior class probability is known. Since $Q_i(y_i)$ is the one-hot vector, we set $S_j := \{(x_i, y_i)|y_{ij} = 1\}$ as the subset of all samples whose labels belong to category $j$. Then we convert Eq. (8) to:

$$\max_{\theta} \sum_{j=1}^K \sum_{i \in S_j} \log p(x_i|y_{ij} = 1, \theta). \quad (10)$$

Following (Wang et al., 2022), we assume that data representations in the embedding space follow a *d-variate von Mises-Fisher (vMF)* distribution whose probabilistic density is given by $f(x|\overline{\mu}_i, \kappa) = c_d(\kappa)e^{\kappa \overline{\mu}_j^{\mathrm{T}} z}$, where $\overline{\mu_i} = \mu_{\mathbf{i}}/\|\mu_{\mathbf{i}}\|$ is the mean direction and $\mu_i$ is the mean center of $S_j$, $\kappa$ is the concentration parameter, and $c_d(\kappa)$ is the normalization factor. We derive Theorem 1.

**Theorem 1.** *Assuming data in the contrastive embedding space follow a d-variate von Mises-Fisher (vMF) distribution $f(x|\overline{\mu}_i, \kappa) = c_d(\kappa)e^{\kappa \overline{\mu}_i^{\mathrm{T}} z}$, then minimizing the expectation of contrastive loss $\mathcal{L}_{cont}(\theta; x, \tau)$ in Eq. (1) also maximizes the likelihood Eq. (10). Formally, we have:*

$$\arg\max_{\theta} \sum_{j=1}^K \sum_{i \in S_j} \log p(x_i|y_{ij} = 1, \theta) = \arg\max_{\theta} \sum_{j=1}^K \sum_{i \in S_j} (\kappa \overline{\mu}_j^{\mathrm{T}} z_i) \geq \arg\min_{\theta} E(\mathcal{L}_{cont}(\theta; x, \tau)). \quad (11)$$

The proof can be found in Appendix A. Theorem 1 indicates that minimizing the expectation of Eq. (1) also maximizes a lower bound of likelihood in Eq. (10). According to (Bottou & Bengio, 1994), by alternating the two steps above, DAST converges to a (perhaps locally) optimal point. We empirically show the clustering effect with t-SNE (Van der Maaten & Hinton, 2008) in Section 5.

**Extensions of DAST with data augmentations.** Thanks to its flexibility, DAST can be readily extended with the existing techniques in advanced SSL and contrastive learning literature. In

particular, for a domain with effective data augmentations (Berthelot et al., 2020; Sohn et al., 2020), DAST can directly utilize these data augmentations to generate the reliable augmented samples for model training. The augmented samples are used to construct the positive and negative sample pairs for the projection head and generate reliable pseudo labels with $p(y|x)$ for the classification head. We introduce the details of DAST extensions in graph and image domains in Appendix D and empirically show its effectiveness in Section 5.

## 5 EXPERIMENTS

### 5.1 EXPERIMENTS IN UNDER-EXPLORED DOMAIN

We first evaluate the performance of DAST in tabular domain where effective data augmentation is not available (Yoon et al., 2020; Somepalli et al., 2021; Bahri et al., 2022). More experiments including ablation study, hyperparameters analysis, quality of the generated pseudo labels, and the training efficiency could be found in Appendix E.

**Datasets.** We select 7 datasets in tabular domain from the OpenML-CC18 benchmark (Bischl et al., 2021). For each dataset, we randomly split it by 8:1:1 to obtain the training, validation, and test sets. For the choices of the datasets, we consider the following aspects: (1) consisting of both numerical and categorical features, (2) only containing numerical or categorical features, (3) performing binary- or multi-classification tasks, and (4) the size of the dataset. Following (Sohn et al., 2020), we perform experiments with lower label ratio (the proportion of labeled training samples to all training data) to empirically demonstrate that DAST shows promise in label-scarce settings. The statistics of the seven datasets as well as the preprocessing details are summarized in Appendix C.

**Comparison methods.** We compared our proposed DAST with the following baselines, which can be divided into three categories. (1) **Domain-agnostic (DA) self-training approaches**: we compare with Π-Model (Rasmus et al., 2015), Pseudo-Label (Lee et al., 2013), and Mean Teacher (Tarvainen & Valpola, 2017) as the domain-agnostic baselines. These approaches do not rely on domain-specific data augmentation. (2) **Domain-specific (DS) self-training approaches**: we compare with VIME (Yoon et al., 2020), a self-training approach which corrupts features in tabular data to construct the augmented samples. Furthermore, we modify Meta Pseudo-Label (MPL) (Pham et al., 2021), UPS (Rizve et al., 2021), CoMatch (Li et al., 2021), SimMatch (Zheng et al., 2022), two advanced self-training approaches and two self-training combined with contrastive learning methods in image domain, to show the performance of costuming the domain-specific methods for tabular data. Specifically, we utilize the data augmentation in VIME to replace the original data augmentations in image domain. (3) **Contrastive learning (CL) approaches**: we compare with two recent advanced contrastive learning approaches, i.e., SCARF (Bahri et al., 2022) and SubTab (Ucar et al., 2021). It is worth mentioning that contrastive learning is also effective for solving SSL problems in tabular domain (Bahri et al., 2022), which first pretrains a model with unlabeled training samples using contrastive loss and then fine-tunes a classifier based on the learned data representations using labeled training samples. Furthermore, we also derive SCMPL (SCARF+MPL) as a baseline which first pretrains the model with SCARF and then fine-tune the classifier with MPL.

**Implantation details.** We use a transformer network as the encoder following (Somepalli et al., 2021), a 2-layer MLP as the projection head, and a 2-layer MLP as the classification head, with ReLU nonlinearity for all layers. As suggested by (Oliver et al., 2018), for fair comparison, we implement all the comparison approaches and perform all experiments using the same codebase. In particular, we use the same network architecture and training protocol, including the optimizer, learning rate, weight decay, and data preprocessing. We tune the hyperparameters in each method with grid search using validation sets. The detailed implementations of the baselines and DAST are presented in Appendix C.. All the experiments were conducted on a Linux server equipped with Intel Xeon 2.10GHz CPUs and NVIDIA GeForce RTX2080Ti GPUs using Pytorch.

**Effectiveness of DAST.** Table 1 shows the performance comparison results of different methods in tabular domain and we have the following important observations. First, our proposed DAST yields the highest test accuracy over 12 out of 14 cases w.r.t. different label ratios and datasets. It achieves 1.05 and 0.89 higher average test accuracy than the second best model with 5% and 10% label ratios, respectively. Second, DAST outperforms both domain-specific self-training and contrastive learning approaches in tabular domain. The reason may lie in the ineffectiveness of the predefined data

Table 1: Overall prediction performance on seven tabular datasets from OpenML (Bischl et al., 2021) with different label ratios. We report the mean AuROC score for binary classification and accuracy for multi-classification with five different seeds. The **bold** value denotes the highest mean score and the second best values are underlined. OOM and Avg are short for out of memory and average result.

| Type | Method | Adult | KDD-cup | CTR-small | Diabetes | Mfeat | Covertype | Solar-flare | Avg. |
|---|---|---|---|---|---|---|---|---|---|
| | | | | label ratio = 5% | | | | | |
| Supervised. | Transformer | 86.43±0.67 | 68.97±0.07 | 60.93±0.41 | 80.75±0.05 | 88.74±0.69 | 88.91±0.28 | 68.94±0.47 | 77.67 |
| CL. | SCARF | 84.72±0.27 | 68.14±0.32 | 61.21±0.14 | 80.54±0.36 | 89.40±0.61 | 88.96±0.49 | 67.41±0.43 | 77.19 |
| | SubTab | 85.19±0.45 | 67.55±0.43 | 60.07±0.21 | 80.09±0.65 | 89.68±0.37 | 88.46±0.62 | 66.23±0.95 | 76.75 |
| | SCMPL | 86.27±0.38 | 68.33±0.81 | 60.69±0.54 | 81.52±0.36 | 90.55±0.34 | 88.78±0.31 | 67.71±0.26 | 77.69 |
| Self-training. (DS) | VIME-Semi | 88.23±0.62 | OOM | 62.45±0.25 | 82.09±0.63 | 90.98±1.17 | OOM | 68.95±0.71 | 78.54 |
| | CoMatch | 87.24±0.44 | 67.32±0.26 | 61.50±0.58 | 81.85±0.33 | 90.58±0.51 | 88.67±0.61 | 69.13±0.34 | 78.04 |
| | SimMatch | 87.92±0.22 | 67.34±0.51 | 61.40±0.40 | 82.13±0.36 | 90.79±0.72 | 88.89±0.53 | 69.13±0.36 | 78.23 |
| | UPS | 87.78±0.23 | 67.46±0.45 | 61.71±0.24 | 82.15±0.35 | 90.93±0.62 | 89.21±0.57 | 69.09±0.69 | 78.34 |
| | MPL | 87.46±0.35 | 68.61±0.45 | 61.97±0.14 | 82.34±0.51 | 91.02±0.73 | 89.18±0.64 | 69.31±0.42 | 78.55 |
| Self-training. (DA) | Π-Model | 86.15±0.14 | 69.06±0.16 | 61.05±0.17 | 81.72±0.68 | 90.02±0.53 | 89.21±0.65 | 69.28±0.39 | 78.07 |
| | Pseudo-Label | 86.92±0.31 | 69.55±0.53 | 60.41±0.24 | 81.44±0.48 | 90.68±0.78 | 89.09±0.52 | 68.92±0.68 | 78.14 |
| | Mean Teacher | 87.75±0.19 | 69.87±0.32 | 61.63±0.19 | 81.95±0.65 | 90.15±0.89 | 89.40±0.41 | 69.16±0.59 | 78.55 |
| | **DAST** | **89.67±0.17** | **70.45±0.48** | **63.82±0.12** | **82.64±0.07** | **91.67±0.62** | **89.45±0.37** | **69.53±0.51** | **79.60** |
| | | | | label ratio = 10% | | | | | |
| Supervised. | Transformer | 87.93±0.14 | 70.52±0.30 | 62.14±0.33 | 81.32±0.14 | 92.17±0.52 | 89.24±0.11 | **70.25±0.22** | 79.08 |
| CL. | SCARF | 85.44±0.21 | 68.91±0.42 | 61.52±0.27 | 82.25±0.61 | 91.57±0.48 | 88.53±0.12 | 69.62±0.37 | 78.26 |
| | SubTab | 86.72±0.36 | 68.51±0.28 | 61.94±0.11 | 82.05±0.45 | 91.42±0.57 | 88.85±0.31 | 69.14±0.67 | 78.37 |
| | SCMPL | 87.32±0.25 | 71.02±0.30 | 62.18±0.17 | 82.81±0.38 | 91.75±0.37 | 89.65±0.23 | 69.56±0.26 | 79.18 |
| Self-training. (DS) | VIME-Semi | 88.96±0.25 | OOM | 63.49±0.55 | 83.94±0.81 | 91.78±0.43 | OOM | 69.28±0.48 | 79.49 |
| | CoMatch | 88.64±0.44 | 70.42±0.26 | 62.64±0.58 | 83.25±0.33 | 92.48±0.51 | 89.63±0.34 | 69.63±0.34 | 79.56 |
| | SimMatch | 89.02±0.36 | 70.61±0.57 | 62.42±0.18 | 83.75±0.43 | 92.57±0.37 | 90.31±0.35 | 69.41±0.42 | 79.72 |
| | UPS | 89.07±0.23 | 69.84±0.34 | 61.25±0.28 | 83.85±0.15 | 93.04±0.27 | 89.72±0.34 | 68.56±0.46 | 79.33 |
| | MPL | 88.24±0.26 | **72.08±0.48** | 62.39±0.29 | 83.25±0.52 | 92.62±0.34 | 89.87±0.23 | 69.72±0.36 | 79.73 |
| Self-training. (DA) | Π-Model | 89.11±0.19 | 71.56±0.66 | 61.80±0.17 | 83.45±0.78 | 91.62±0.43 | 89.62±0.37 | 69.98±0.31 | 79.61 |
| | Pseudo-Label | 88.68±0.43 | 69.12±0.76 | 61.96±0.36 | 82.98±0.54 | 91.16±0.42 | 89.68±0.49 | 68.62±0.52 | 78.88 |
| | Mean Teacher | 88.47±0.17 | 71.28±0.29 | 62.20±0.29 | 83.39±0.72 | 92.17±0.81 | 89.54±0.36 | 68.96±0.44 | 79.43 |
| | **DAST** | **89.95±0.09** | 71.75±0.15 | **64.26±0.09** | **84.93±0.02** | **93.23±0.04** | **90.47±0.14** | 69.90±0.05 | **80.62** |

(a) Original features     (b) MPL     (c) DAST

Figure 2: T-SNE visualization of the data representations on Mfeat dataset.

augmentations in tabular domain since randomly corrupting features to generate augmented samples would easily change data semantics and thus inject noise during model training. Our proposed DAST liberates the dependence on data augmentation and shows the superiority for tabular domain without useful and insightful data augmentation. Third, SCMPL performs worse than MPL in all the cases. This indicates that simply combining contrastive learning and self-training without effective data augmentation may inject more noisy data that hurt model's performance. In contrast, DAST assigns both reliable contrastive pairs and pseudo labels that benefits the model performance.

**Learned distinguishable data representations of DAST.** We visualize the data representations (on Mfeat dataset with 10% label ratio) using t-SNE (Van der Maaten & Hinton, 2008) in Figure 2 to show that DAST can learn distinguishable representations. Different colors denote the ground-truth class labels. We show the t-SNE embeddings of (1) original numerical feature (2) data representations learned by the best baseline MPL, and (3) data representations learned by DAST. We observe that the representations of original numerical features are indistinguishable. MPL can learn distinguishable data representations, but a few examples in different classes are still overlapped (e.g., samples in yellow and purple). DAST produces distinguishable data representations that lead to a better class separation, which confirms the effectiveness of DAST in learning high-quality data representations.

## 5.2 EXPERIMENTS IN OTHER DOMAINS

We conduct the experiments in graph domain which has no universal data augmentation that is effective for different datasets (You et al., 2020; 2021; Lee et al., 2022), showing that DAST can combine with less effective data augmentations to consistently improve the model performance.

Moreover, we also conduct the experiments in image domain to show that existing effective data augmentations can be directly used as add-ons for DAST to further boost the model performance. The complete experimental settings could be found in Appendix C.

Table 2: Overall prediction performance on five graph datasets from TUDataset. The full results are listed in Appendix D.

| Type | Method | NCI1 | DD | PROTEINS | MUTAG | IMDB-B |
|---|---|---|---|---|---|---|
| Supervised. | ResGCN | 73.21 | 73.78 | 70.35 | 87.45 | 71.16 |
| CL. | ContextPred | 73.11 | 74.82 | 70.29 | 89.69 | 72.30 |
| | InfoGraph | 74.21 | 75.94 | 71.69 | 90.33 | 74.91 |
| | GraphCL | 75.86 | 75.84 | **73.75** | 89.80 | 75.26 |
| | JOAO | 76.14 | 75.52 | 72.98 | 90.67 | 75.66 |
| Self-training. (DA) | Π-Model | 71.82 | 72.86 | 69.74 | 88.68 | 72.26 |
| | Pseudo-Label | 72.31 | 71.52 | 69.65 | 88.91 | 72.18 |
| | Mean Teacher | 73.51 | 73.54 | 70.83 | 89.60 | 71.94 |
| | DAST | 75.27 | 73.92 | 71.01 | 89.72 | 72.02 |
| Self-training. (DS) | MPL | 73.28 | 73.45 | 70.62 | 89.44 | 71.64 |
| | InfoGraph-Semi | 75.77 | 75.11 | 73.27 | 90.99 | 74.82 |
| | DAST+ | **78.76** | **76.19** | 72.98 | **92.22** | **75.80** |

Table 3: Overall prediction performance on three image datasets. The full results are represented in Appendix D.

| Type | Method | CIFAR-10 | CIFAR-100 | STL-10 |
|---|---|---|---|---|
| Supervised. | WRN | 78.12 | 39.83 | 71.34 |
| CL. | MOCO | 78.14 | 52.82 | 81.76 |
| | SimCLR | 79.25 | 54.61 | 80.29 |
| Self-training. (DA) | Π-Model | 72.19 | 38.29 | 72.88 |
| | Pseudo-Label | 75.78 | 36.54 | 70.25 |
| | Mean Teacher | 80.79 | 40.21 | 74.56 |
| | DAST | 81.32 | 41.09 | 74.91 |
| Self-training. (DS) | ReMixMatch | 92.31 | 67.85 | 93.17 |
| | FixMatch | 92.92 | 67.42 | 93.25 |
| | MPL | 93.29 | 68.55 | 93.21 |
| | SimMatch | 93.42 | 68.95 | 93.37 |
| | DAST+ | **93.74** | **69.95** | **93.81** |

**Datasets.** We present the results of applying DAST to graph classification problems using five well-known datasets from the benchmark TUDataset (Morris et al., 2020) and evaluate the performance of DAST on image classification tasks using three benchmark image datasets.

**Comparisons methods and experimental settings.** To evaluate the performance of DAST in graph and image domains, we consider the state-of-the-art domain-specific self-training and the contrastive learning approaches. Specifically, we compare DAST with ContextPred (Hu et al., 2019), InfroGraph (Sun et al., 2019), GraphCL (You et al., 2020), and JOAO (You et al., 2021) in graph domain. For image domain, we consider two additional advanced self-training approaches ReMixMatch (Berthelot et al., 2020) and FixMatch (Sohn et al., 2020), and two contrastive learning approaches MOCO (He et al., 2020) and SimCLR (Chen et al., 2020a). All approaches use the same backbone model,where we adopt ResGCN (Chen et al., 2019) for graph classification and WideResNet (Zagoruyko & Komodakis, 2016) for image classification. We extend DAST to DAST+ with domain-specific data augmentations to verify the scalability of DAST and the details of how to apply existing data augmentations with DAST could be found in Appendix D.

**Results.** Table 2 shows the comparison results on 5 graph classification datasets with 10% label ratio and Table 3 shows the comparison results in image domain on CIFAR-10, CIFAR-100 and STL-10 with 1000, 2500 and 1000 labeled samples, respectively. We have the following observations. (1) DAST consistently outperforms other DA self-training approaches, which again verifies the superiority of DAST without using data augmentation. (2) In graph domain, DAST closely matches the performance of some DS self-training approaches, i.e., ContextPred and InfoGraph. These results are appealing since DAST uses no domain knowledge. (3) DAST+ in both graph and image domains achieve the highest average test accuracy, which implies that DAST can combine with existing data augmentations to further boost the model performance. Moreover, while DAST+ utilizes the less effective data augmentation NodeDrop in graph domain, it can consistently outperforms JOAO which adaptively selects optimal data augmentations for each dataset. This demonstrates that DAST can select reliable augmented samples during model training to improve the model performance.

## 6 CONCLUSION

In this paper, we introduce a generic self-training framework named DAST which is applicable to domains without effective data augmentations. DAST decouples the generation and utilization of pseudo labels by incorporating the classification module with a contrastive learning module. It employs a novel two-way pseudo label generation strategy that facilitates each module to generate reliable pseudo labels for the other without the reliance on domain-specific data augmentation. Theoretically, we have proved that the contrastive learning and the self-training in DAST are mutually beneficial from the EM-algorithm perspective. Extensive experiments on real datasets in tabular, image and graph domains verify the effectiveness of DAST in SSL tasks compared with the advanced self-training approaches. We also empirically show that existing data augmentations can be used as add-ons to DAST to further boost the model performance.

**Ethics Statement:** Our proposed DAST is domain-agnostic and can be applied to domains without any prior knowledge. Besides, our method does not generate new samples during model training or require human subjects. To the best of our knowledge, DAST has no potential harmful insights and negative social impacts.

**Reproducibility:** We clarify the assumptions in Section 4 and provide the complete proofs of Theorems in Appendix A.The statistics of datasets, the data processing, and the details of the experimental settings are described in Appendix C. Our code could be found in the `https://github.com/anonymous202301/DAST`.

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

# Appendix

In this appendix, we first give the complete proof of Theorem 1 (Appendix A) and provide the pseudo-code of the DAST framework (Appendix B). The details of the experimental settings are presented in Appendix C. We report the complete experimental results in graph and image domains (Appendix D), the ablation study of DAST (Appendix E), and the additional experiments in tabular domain (Appendix F). Finally, we provide the complexity analysis in Appendix G and additional discussions about the difference between DAST and domain adaption approaches and the limitations of DAST in Appendix H.

## A THEORETICAL ANALYSIS

We denote $S_j = \{(x_i, y_i)|y_{ij} = 1\}$ as the subset of all samples in the training dataset whose labels belong to category $j$. We assume the data representations follow a *d-variate von Mises-Fisher (vMF)* distribution whose probabilistic density is given by $f(x|\overline{\mu}_i, \kappa) = c_d(\kappa)e^{\kappa \overline{\mu}_i^\top z}$, where $\overline{\mu}_i = \mu_i/\|\mu_i\|$ is the mean direction, $\kappa$ is the concentration parameter, and $c_d(\kappa)$ is the normalization factor.

**Theorem 1.** *Assume data in the contrastive embedding space follow a d-variate von Mises-Fisher (vMF) distribution $f(x|\overline{\mu}_i, \kappa) = c_d(\kappa)e^{\kappa \overline{\mu}_i^\top z}$, then minimizing the expectation of contrastive loss $\mathcal{L}_{cont}(\theta; x, \tau)$ in Eq. (1) also maximizes the likelihood in Eq. (11). Formally, we have:*

$$\arg\max_\theta \sum_{j=1}^K \sum_{i \in S_j} \log p(x_i|y_{ij} = 1, \theta) = \arg\max_\theta \sum_{j=1}^K \sum_{i \in S_j} (\kappa \overline{\mu}_j^\top z_i) \geq \arg\min_\theta \mathrm{E}(\mathcal{L}_{cont}(\theta; x, \tau)).$$

*Proof.* Since the data in the embedding space follow a *d-variate von Mises-Fisher (vMF)* distribution, we can convert Eq.(11) with data in different prediction categories $S_j$ as:

$$\arg\max_\theta \sum_{j=1}^K \sum_{i \in S_j} \log p(x_i|y_{ij} = 1, \theta) = \arg\max_\theta \sum_{j=1}^K \sum_{i \in S_j} \log(c_d(\kappa)e^{\kappa \overline{\mu}_j^\top z}) = \arg\max_\theta \sum_{j=1}^K \sum_{i \in S_j} (\kappa \overline{\mu}_j^\top z_i). \tag{13}$$

We ignore the constant factor $\log(c_d(\kappa))$ in the last equality since it is independent with model parameters $\theta$. Denote $D = D_L \cup D_U$ as the whole training set with $N + M$ training samples. In each training step, if all the labeled and unlabeled samples are accessible, the expectation of supervised contrastive loss in Eq.(1) is defined as:

$$\mathrm{E}(\mathcal{L}_{cont}(\phi, h_p; x, \tau)) = \frac{1}{N+M} \sum_{x \in D} \left(-\frac{1}{|E(x)|} \sum_{e \in E(x)} \log \frac{\exp(z^\top z_e/\tau)}{\sum_{a \in A(x)} \exp(z^\top z_a/\tau)}\right)$$

$$= \frac{1}{N+M} \sum_{x \in D} \left(-\frac{1}{|E(x)|} \sum_{e \in E(x)} (z^\top z_e/\tau)\right) + \frac{1}{N+M} \sum_{x \in D} \left(\frac{1}{|E(x)|} \sum_{a \in A(x)} (z^\top z_a/\tau)\right) \tag{14}$$

Here we focus on the first term of Eq.(14). According to (Wang & Isola, 2020), the first term is used to optimize the intraclass covariance in the embedding space to pose a tight clustering effect, which is often dubbed as *alignment* term. Note that we use the supervised contrastive loss to learn the data representations. If we consider an ideal setting which all the pseudo labels $\hat{y}^c$ produced by the classification head is accurate. Given a sample $x_i$ with the ground-truth label $y_{ij} = 1$, the subset of selected positive contrastive pairs is equivalent to the subset of data which contains the same label, i.e., $E(x) = S_j$. Similar to the Eq.(11), we could convert the first term of Eq.(14) as:

$$\arg\min_\theta \frac{1}{N+M} \sum_{x \in D} \left(-\frac{1}{|E(x)|} \sum_{e \in E(x)} (z^\top z_e/\tau)\right) = \arg\min_\theta \sum_{j=1}^K \sum_{i \in S_j} \left(-\frac{1}{|S_j|} \sum_{e \in S_j} (z_i^\top z_e/\tau)\right)$$

$$= \arg\min_\theta \sum_{j=1}^K \sum_{i \in S_j} (-\mu_j^\top z_i/\tau)) \tag{15}$$

$$= \arg\max_\theta \sum_{j=1}^K \sum_{i \in S_j} (\mu_j^\top z_i/\tau)$$

Recall that $\overline{\mu_i} = \mu_{\mathbf{i}}/\|\mu_{\mathbf{i}}\|$, thus we have the following equation:

$$
\begin{aligned}
\arg\max_\theta \sum_{j=1}^K \sum_{i\in S_j} (\mu_{\mathbf{j}}^\top z_i/\tau)) = \arg\max_\theta \sum_{j=1}^K (\|\mu_{\mathbf{j}}\|^2/\tau)) & \\
\leq \arg\max_\theta \sum_{j=1}^K (\|\mu_{\mathbf{j}}\|)) & \\
= \arg\max_\theta \sum_{j=1}^K \sum_{i\in S_j} (\kappa\overline{\mu}_j^\top z_i). &
\end{aligned}
\tag{16}
$$

Eq.(16) holds since the data embeddings are distributed on the hypersphere $R^{d-1}$ and thus $\|\mu_{\mathbf{i}}\| \in [0,1]$. Putting the second term in Eq.(14 together, minimize the contrastive loss also maximizes a lower bound of the likelihood in Eq.(11) since we have:

$$
\begin{aligned}
\arg\max_\theta \sum_{j=1}^K \sum_{i\in S_j} \log p(x_i|y_{ij}=1,\theta) \geq \arg\min_\theta \sum_{x\in D} (-\frac{1}{|E(x)|} \sum_{e\in E(x)} (z^\top z_e/\tau)) & \\
\geq \arg\min_\theta \mathrm{E}(\mathcal{L}_{cont}(\theta;x,\tau)). &
\end{aligned}
\tag{17}
$$

Therefore, we have proved that minimizing Eq.(1 is equivalent to maximizing a lower bound of the likelihood in Eq.(11). $\qquad\square$

## B  PSEUDO-CODE OF DAST

We present the pseudo-code of our DAST framework in Algorithm 1.

---

**Algorithm 1:** Pseudo-code of DAST (one epoch)

---

**Input:** Labeled training dataset $D_L = \{(x_i, y_i)\}$, unlabeled training dataset $D_U = \{x_i\}$, encoder $\phi(x)$, classification head $h_c$, projection head $h_p$, iteration $N$, threshold $\alpha$, temperature $\tau$, weight of unsupervised loss $\lambda$.

1 **for** $iter \in [1, 2, \ldots, N]$ **do**
2 $\quad$ sample mini-batch $\mathcal{D}_L$ and $\mathcal{D}_U$ from labeled set $D_L$ and unlabeled set $D_L$;
3 $\quad$ $\mathcal{L}_{cls}(\phi, h_c; \mathcal{D}_L) = \frac{1}{|\mathcal{D}_L|} \sum_{i\in\mathcal{D}_L} H(y_i, h_c \circ \phi(x_i));$ $\qquad\qquad$ */ Cross-entropy loss in $\mathcal{D}_L$*
4 $\quad$ $E_L(x_i) = \{x_e \in \mathcal{D}_L|y_e = y_i\}, A_L(x_i) = \mathcal{D}_L \setminus E_L(x_i);$ */ Positive & negative set generation in $\mathcal{D}_L$*
5 $\quad$ $\mathcal{L}_{cont}(\phi, h_p; \mathcal{D}_L, \tau) = \frac{1}{|\mathcal{D}_L|} \sum_{i\in\mathcal{D}_L} \frac{-1}{|E_L(x_i)|} \sum_{e\in E_L(x_i)} \log \frac{\exp(z_i^\top z_e/\tau)}{\sum_{a\in A_L(x)} \exp(z_i^\top z_a/\tau)};$
$\qquad$ */ Supervised contrastive loss in $\mathcal{D}_L$*
6 $\quad$ **for** $x_i \in \mathcal{D}_U$ **do**
7 $\quad\quad$ **for** $x_j \in \mathcal{D}_U$ **do**
8 $\quad\quad\quad$ $g_{ij} \leftarrow$ calculate the similarity $sim(z_i, z_j)$ with Eq. (5); $\qquad$ */ Construct neighbor graph*
9 $\quad\quad$ **end**
10 $\quad\quad$ $\hat{y}_i^c \leftarrow \mathbb{1}\{\arg\max_{j\in\mathcal{Y}}(h_c \circ \phi)^j(x_i) > \alpha\};$ $\quad$ */ Generate pseudo label with the classification head*
11 $\quad$ **end**
12 $\quad$ $\mathcal{W} = D^{-1/2}(G + G^\mathrm{T})D^{-1/2};$ $\qquad\qquad\qquad\qquad$ */ Symmetrically normalized neighbor graph*
13 $\quad$ generate the pseudo labels $\hat{y}_i^p$ for $\mathcal{D}_U$ using Eq. (6), Eq. (7), and Eq. (8);
14 $\quad$ $\mathcal{L}_{cls}(\phi, h_c; \mathcal{D}_U) = \frac{1}{|\mathcal{D}_U|} \sum_{i\in\mathcal{D}_U} H(\hat{y}_i^p, h_c \circ \phi(x_i));$ $\qquad\qquad$ */ Cross-entropy loss in $\mathcal{D}_U$*
15 $\quad$ $E_U(x_i) = \{x_e \in \mathcal{D}_U|\hat{y}_e^c = \hat{y}_i^c\}, A_U(x_i) = \mathcal{D}_U \setminus E_U(x_i);$ */ Positive & negative set generation in $\mathcal{D}_U$*
16 $\quad$ $\mathcal{L}_{cont}(\phi, h_p; \mathcal{D}_U, \tau) = \frac{1}{|\mathcal{D}_U|} \sum_{i\in\mathcal{D}_U} \frac{-1}{|E_U(x_i)|} \sum_{e\in E_U(x_i)} \log \frac{\exp(z_i^\top z_e/\tau)}{\sum_{a\in A_U(x)} \exp(z_i^\top z_a/\tau)};$
$\qquad$ */ Supervised contrastive loss in $\mathcal{D}_U$*
17 $\quad$ minimize the total training loss:
18 $\quad$ $\mathcal{L} = \mathcal{L}_{cls}(\phi, h_c; \mathcal{D}_L) + \mathcal{L}_{cont}(\phi, h_p; \mathcal{D}_L, \tau) + \lambda(\mathcal{L}_{cls}(\phi, h_c; \mathcal{D}_U) + \mathcal{L}_{cont}(\phi, h_p; \mathcal{D}_U, \tau));$
19 **end**

---

## C  EXPERIMENTAL SETTINGS

Table 4: The statistics of the seven tabular datasets from the OpenML-CC18 (Bischl et al., 2021).

| Property \Dataset | Adult | KDD-cup | CTR-small | Diabetes | Mfeat | Covertype | Solar-flare |
|---|---|---|---|---|---|---|---|
| # Instances | 48842 | 50000 | 39948 | 768 | 2000 | 581012 | 1066 |
| # Numerical Features | 6 | 192 | 5 | 8 | 7 | 10 | 0 |
| # Categorical Features | 9 | 39 | 7 | 0 | 0 | 45 | 13 |
| # Classes | 2 | 2 | 2 | 2 | 10 | 7 | 6 |

Table 5: The statistics of the five graph datasets in diverse categories from the TUDataset (Morris et al., 2020).

| Property \Dataset | NCI1 | DD | PROTEINS | MUTAG | IMDB-B |
|---|---|---|---|---|---|
| Category | Molecules | Bioinformatics | Bioinformatics | Molecules | Social networks |
| # Graph Count | 4110 | 1178 | 1113 | 188 | 1000 |
| # Average Node | 29.87 | 284.32 | 39.06 | 17.93 | 19.77 |
| # Average Degree | 1.08 | 715.66 | 1.86 | 19.79 | 96.53 |

## C.1 DATASETS DESCRIPTION

**Tabular datasets.** Table 4 shows the statistics of the datasets. We select 7 tabular datasets from the OpenML-CC18 benchmark (Bischl et al., 2021). The selected datasets consist of (1) both numerical and categorical features, (2) only numerical features, (3) only categorical features, and (4) binary and multiple classes. As for the data preprocessing, following (Somepalli et al., 2021), we preprocess each dataset by Z-normalizing all numerical features and by label-encoding all categorical features before data is passed to the encoder. Each feature (or column) has a different missing value token to account for missing data. We randomly split the dataset by 8:1:1 to obtain the training, validation, and test sets. All the datasets could be found in `https://www.openml.org/search?type= data&status=active`.

**Graph datasets.** We use datasets of diverse nature from the benchmark TUDataset (Morris et al., 2020), including graph data for small molecules, bin-informatics and relation social networks. The statistics of the datasets are shown in Table 5. Specifically, the datasets are selected from diverse nature with the count of graphs, average nodes and average degrees in a border range. We randomly split the dataset by 8:1:1 to obtain the training, validation, and test sets. All the datasets could be found in `https://chrsmrrs.github.io/datasets/docs/datasets/`.

**Image datasets.** We present the performance of DAST on three SSL image classification benchmarks, including CIFAR-10[1], CIFAR-100[1] and STL-10[2]. Specifically, the CIFAR-10 dataset consists of 60000 images in 10 classes with $32 \times 32$ pixels where 50000 images are used for model training and 10000 images for testing. During model training, we further split 20% of training data as validation set and use the provided test set data to evaluate the model performance. The CIFAR-100 dataset is similar to CIFAR-10 which contains 50000 training images and 10000 test images in 100 classes with 600 images per class. We also split 20% of training data as validation set for model tuning and use the provided test set for testing. The STL-10 dataset has predefined 5000 labeled training images (5 of the predefined folds where each fold contain 1,000 labeled images), 10000 unlabeled training samples, and 8000 test images with $96 \times 96$ pixels from 10 classes. There exists out-of-distribution images in the unlabeled set, making it a more realistic and challenging test of SSL performance. During model training, we select one predefined data fold (1000 images) as labeled training set, 10000 unlabeled training data as unlabeled training set, another predefined data fold as validation set, and 8000 test images as test set.

## C.2 EXPERIMENTAL SETTINGS

**Tabular data.** We use a transformer network as the encoder following (Somepalli et al., 2021), a 2-layer MLP as the projection head, and a 2-layer MLP as the classification head, with ReLU nonlinearity for all layers. For all the methods, we set batchsize to 512 (256 for labeled data, 256

---

[1]`https://www.cs.toronto.edu/~kriz/cifar.html`
[2]`https://cs.stanford.edu/~acoates/stl10/`

for unlabeled data) and apply AdamW (Loshchilov & Hutter, 2019) optimizer with the learning rate in $\{10^{-3}, 10^{-4}\}$ and the weight decay of $5 \times 10^{-4}$. The domain-agnostic self-training approaches ($\Pi$-Model Rasmus et al. (2015), Pseudo-Label (Lee et al., 2013), Mean Teacher Tarvainen & Valpola (2017), and DAST) are trained with 120 epochs without any domain knowledge. We apply domain-specific self-training approach VIME (Yoon et al., 2020), UPS (Rizve et al., 2021), MPL (Pham et al., 2021), CoMatch (Li et al., 2021) and SimMatch (Zheng et al., 2022) with 120 training epochs using the data augmentations in the VIME. Following the standard experimental setup in contrastive learning (Chen et al., 2020a; He et al., 2020; Bahri et al., 2022), SCARF (Bahri et al., 2022) and SubTab (Ucar et al., 2021) are first pretrained with 100 epochs using contrastive loss and then fine-tuned on a classifier with 50 epochs. Here we apply the data augmentations in the original papers to construct the contrastive pairs. The proposed SCMPL (the combination of SCARF and MPL) is first pretrained with 100 epochs following Scarf and then fine-tuned using MPL with 50 epochs.

The hyperparameters of each method are tuned with validation sets using grid search.

- **$\Pi$-Model.** We search unlabeled loss weight $\lambda$ in $\{0.1, 0.3, 1.0, 3.0\}$, warm-up epochs of unlabeled loss in $\{20, 30, 40\}$.

- **Pseudo-Label**. We search the confidence threshold $\alpha$ in $\{0.7, 0.8, 0.9, 0.95\}$, and the weight of unlabeled loss $\lambda$ in $\{0.1, 0.3, 1.0, 3.0\}$.

- **Mean Teacher.** We fix the exponential moving average hyperparameter to $0.999$. We search unlabeled loss weight $\lambda$ in $\{0.1, 0.3, 1.0, 3.0\}$, warm-up epochs of unlabeled loss in $\{20, 30, 40\}$.

- **SCARF.** To pretrain SCARF, we follow the best hyperparameters suggested in the original paper. Using the learned representations, we fine-tune a classification head which has the same architecture as the classification head in DAST with the learning rate in $\{10^{-3}, 10^{-4}, 10^{-5}\}$.

- **SubTab.** The same as SCARF.

- **VIME.** We follow the hyperparameter setting suggested in the original paper. Specifically, we search $p_m \in [0.1, 0.9]$ as the proportion of masked and corrupted features, $\beta \in [0.1, 10]$ to make the trade-off between supervised loss and unsupervised loss, and the number of the augmented samples $K$ in $\{2, 3, 5, 10, 15, 20\}$.

- **CoMatch.** We follow the original paper to set the decay rate to $0.0005$, the confidence threshold $\alpha$ to $0.95$, and the weight of unlabeled loss $\lambda$ to $1.0$. We follow the data augmentations in VIME to costume CoMatch for tabular data.

- **SimMatch.** We search the confidence threshold $\alpha$ in $\{0.7, 0.8, 0.9, 0.95\}$, the weight of unlabeled loss $\lambda_u$ in $\{0.1, 0.3, 1.0, 3.0\}$ and the weight of consistency regularization loss $\lambda_{in}$ in $\{0.1, 0.3, 1.0, 3.0\}$. We follow the data augmentations in VIME to costume SimMatch for tabular data.

- **MPL.** We set the decay rate to $0.999$ and search the confidence threshold $\alpha$ in $\{0.7, 0.8, 0.9, 0.95\}$ and the weight of unlabeled loss $\lambda$ in $\{0.1, 0.3, 1.0, 3.0\}$. We follow the data augmentations in VIME to costume MPL for tabular data.

- **UPS.** We search the confidence threshold $\alpha$ in $\{0.7, 0.8, 0.9, 0.95\}$ and uncertainty thresholds $\kappa$ in $\{0.005, 0.05, 0.5\}$. We set the pseudo-labeling iterations to 10 following the original paper. We utilize the data augmentation in VIME to costume UPS for tabular data.

- **SCMPL.** We use the same setting as SCARF for pretraining and the same setting as MPL for fine-tuning.

- **DAST.** We set the temperature $\tau$ to $0.07$ in Eq. (1) and the confidence threshold $\alpha$ to $0.95$ in Eq. (4) by default. We construct the neighbor graph with the number of selected neighbors $k = 50$ in Eq. (5) and update the graph at every 5 epochs, i.e., $T = 5$. The weight of trade-off parameter $\beta$ in Eq. (8) is searched in $\{0.1, 0.3, 0.5, 1.0\}$. We set the weight of unlabeled loss $\lambda = 0$ in Eq. (3) before warm-up epochs $T_{warm}$ and $\lambda = 1$ for the remaining epochs where $T_{warm} \in \{0, 10, 20, 30, 40\}$.

It is worth mentioning that all the domain-agnostic SSL approaches including $\Pi$-Model (Rasmus et al., 2015), Pseudo-Label (Lee et al., 2013), Mean Teacher (Tarvainen & Valpola, 2017) and DAST,

utilize the same hyperparameter settings in all the domains. We omit redundant description in the rest of paper.

Although there are a few hyperparameters in DAST, we only adjust the warm-up epochs $T_{warm}$ and $\beta$ in our experiments to tune the model performance. The other hyperparameters such as confidence threshold and neighbor numbers are fixed to the default values which are also used in the existing self-training methods (Berthelot et al., 2020; Sohn et al., 2020) and graph transductive learning (Chandra & Kokkinos, 2016) for all the experiments.

**Graph data.** We adopt ResGCN (Chen et al., 2019) with 5 layers and 128 hidden dimensions as the encoder following (You et al., 2021), a 2-layer MLP as the projection head, and a 2-layer MLP as the classification head, with ReLU nonlinearity for all layers. For all the methods in graph domain, we set batchsize to 512 (256 for labeled data, 256 for unlabeled data) and apply Adam optimizer with the learning rate of $10^{-3}$ and the weight decay of $5 \times 10^{-4}$. The domain-agnostic self-training approaches (Π-Model (Rasmus et al., 2015), Pseudo-Label (Lee et al., 2013), Mean Teacher (Tarvainen & Valpola, 2017), and DAST) are trained with 100 epochs. We train the domain-specific self-training approach InfoGraph-semi (Sun et al., 2019) and MPL (Pham et al., 2021) with 100 epochs using the predefined subgraph augmentation. The contrastive learning approaches ContextPred (Hu et al., 2019), InfroGraph (Sun et al., 2019), GraphCL (You et al., 2020), and JOAO (You et al., 2021) are first pretrained with 100 epochs using contrastive loss and then fine-tuned with 50 epochs. Here we follow (You et al., 2021) to select the data augmentations in {NodeDrop, Subgraph, EdgePert, AttrMask} for constructing the contrastive pairs.

The hyperparameter settings of domain-specific methods in graph domain are introduced below:

- **ContextPred.** Following the original paper, we define the context graph of a selected node $v$ as the surrounding graph structure that is between $r_1$- and $r_2$-hop from the node $v$. We search the inner radius $r_1$ in $\{1, 2, 3\}$ and the outer radius $r_2$ in $\{2, 3, 4\}$.

- **InfoGraph.** Following the original paper, we utilize LIBSVM (Chang & Lin, 2011) in the fine-tuning stage with the regularization term $C$ selected from $\{10^{-3}, 10^{-2}, ..., 10^3\}$.

- **GraphCL.** We adopt the augmentation in GraphCL with the default augmentation strength of 0.2. We use the hyperparameters suggested in the original paper for both pretraining and fine-tuning stages.

- **JOAO.** Following the original paper, we set the augmentation strength to 0.2 and search the hyperparameter $\gamma$ in $\{0.01, 0.1, 1\}$ to control the influence of prior distribution.

- **MPL.** We set the decay rate to 0.999 and search the confidence threshold $\alpha$ in $\{0.7, 0.8, 0.9, 0.95\}$ and the weight of unlabeled loss $\lambda$ in $\{0.1, 0.3, 1.0, 3.0\}$. We select the data augmentations in {NodeDrop, Subgraph, EdgePert, AttrMask} to costume MPL for each graph dataset.

- **InfoGraph-semi.** Following the original paper, we set the number of set2set computations to 3 and search the hyperparameter $\lambda$ in $\{10^{-3}, 10^{-4}, 10^{-5}\}$ to control the relative weight between the supervised and unsupervised loss.

**Image data.** We closely follow the experimental settings in FixMatch (Sohn et al., 2020), including the backbone model, dataset settings, optimizer and data augmentations. Specifically, we use a WideResNet(WRN)-28-2 (Zagoruyko & Komodakis, 2016) with 1.5M parameters, WRN-28-8 and WRN-37-2 as the encoders for CIFAR-10, CIFAR-100, and STL-10, respectively. We implement two 3-layer MLPs as the projection head and the classification head, respectively. We set batchsize to 1024 (128 labeled examples and 896 unlabeled samples) and apply SGD optimizer with the learning rate of 0.03, weight decay of $5 \times 10^{-4}$, and the cosine learning rate decay. We train the domain-agnostic self-training approaches (Π-Model (Rasmus et al., 2015), Pseudo-Label (Lee et al., 2013), Mean Teacher (Tarvainen & Valpola, 2017), and DAST) with 150 epochs without any domain knowledge. The domain-specific self-training approaches ReMixMatch (Berthelot et al., 2020), MPL (Pham et al., 2021), FixMatch (Sohn et al., 2020) and SimMatch (Zheng et al., 2022) are trained with 150 epochs using the predefined weak data augmentations (e.g., image flip-and-shift) and strong data augmentations (e.g., Cutout and AutoAugment (Cubuk et al., 2020)). As for the contrastive learning approaches MOCO (He et al., 2020) and SimCLR (Chen et al., 2020a), we first pretrain the model with 100 epochs using contrastive loss and then fine-tune a classifier with the learned data

embeddings with 50 epochs. Following (Chen et al., 2020a), we use image crop and flip to construct the contrastive pairs in the pretraining stage.

We now introduce the hyperparameter settings of domain-specific methods in image domain below:

- **MOCO.** For pre-training in MOCO, similar to (Chen et al., 2020a), we use random cropping with random left-to-right flipping as data augmentation. We follow the best hyperparameters suggested in the original paper. As for the fine-tuning stage, we use the classification head which has the same architecture as that in DAST with the learning rate chosen from $\{0.03, 0.05, 0.1\}$.

- **SimCLR.** The same as MOCO.

- **MPL.** We set the decay rate to 0.999 and search the confidence threshold $\alpha$ in $\{0.7, 0.8, 0.9, 0.95\}$ and the weight of unlabeled loss $\lambda$ in $\{0.1, 0.3, 1.0, 3.0\}$. We follow the data augmentation in original paper for images.

- **ReMixMatch.** We set the exponential decay rate to 0.99 and search the confidence threshold $\alpha$ in $\{0.7, 0.8, 0.9, 0.95\}$. The weight of unlabeled loss $\lambda$ is searched in $\{0.1, 0.3, 1.0, 3.0\}$.

- **FixMatch.** Following the original paper, we set the confidence threshold $\tau$ to 0.95 for selecting pseudo labels and the ratio $\mu$ to 7 which determines the relative sizes of unlabeled and labeled training data. The weight of unlabeled loss $\lambda$ is searched in $\{0.1, 0.3, 1.0, 3.0\}$.

- **SimMatch.** We search the confidence threshold $\alpha$ in $\{0.7, 0.8, 0.9, 0.95\}$, the weight of unlabeled loss $\lambda_u$ in $\{0.1, 0.3, 1.0, 3.0\}$ and the weight of consistency regularization loss $\lambda_{in}$ in $\{0.1, 0.3, 1.0, 3.0\}$. We follow the data augmentations used in original paper.

## D    COMPLETE EXPERIMENTAL RESULTS IN GRAPH AND IMAGE DOMAINS

### D.1    GRAPH DOMAIN

**Extension of DAST in graph domain.** As we mentioned in the main body of the paper, DAST can be readily extended to DAST+ using existing data augmentations. Specifically, in graph domain, we first utilize the data augmentation NodeDrop in (You et al., 2020) to generate the augmented samples by randomly discarding certain portion of vertices along with their connections using a default i.i.d. uniform distribution. Following the original paper, we set the discard portion to 0.2 and expect it will not affect the data label. We then produce the pseudo labels using the model predictions on the augmented samples for computing $\mathcal{L}_{cls}$ and perform the supervised contrastive learning with the augmented samples for $\mathcal{L}_{cont}$. Finally, we train DAST+ with the sum loss $\mathcal{L}$ in Eq. (3) using the same experimental setting as DAST.

**Results in graph domain.** Table 6 provides the complete comparison results for graph classification. We have the following important observations. (1) DAST outperforms all the domain-agnostic self-training approaches by a large margin, which again verifies the superiority of DAST without using data augmentation. Moreover, DAST closely matches the performance of some domain-specific approaches, i.e., ContextPred and InfoGraph. These results are particularly appealing since DAST uses no domain knowledge during model training. (2) DAST+ yields the highest average rank in five graph datasets, achieving 0.85 and 1.01 higher average test accuracy than the second best method with 5% and 10% label ratios, respectively. This demonstrates that DAST can combine with existing data augmentation to further boost the model performance. (3) It is worth mentioning that the data augmentation NodeDrop is ineffective for NCI1 and MUTAG datasets. It has been empirically proved that simply training the model with every possible augmentation pairs from NodeDrop would degenerate the model performance (You et al., 2021; Yue et al., 2022). DAST+ achieves the highest accuracy score in these two datasets by only utilizing the reliable pseudo labels and contrastive pairs for model training, leading to better model performance.

### D.2    IMAGE DOMAIN

**Extension of DAST in image domain.** We follow the data augmentation in FixMatch (Sohn et al., 2020) to extend DAST in image domain. Specifically, we utilize the image crop and flip as the weak data augmentations while Cutout (DeVries & Taylor, 2017) and RandAugment (Cubuk et al., 2020)

Table 6: Overall prediction performance on five graph datasets from TUDataset Morris et al. (2020) with different label ratios. We report the mean test accuracy with five different seeds. The bold denotes the highest mean score. A.R. and lr are short for average rank and label ratio, respectively.

| Type | Method | NCI1 | | DD | | PROTEINS | | MUTAG | | IMDB-B | | A.R. |
|---|---|---|---|---|---|---|---|---|---|---|---|---|
| | | lr=5% | lr=10% | lr=5% | lr=10% | lr=5% | lr=10% | lr=5% | lr=10% | lr=5% | lr=10% | |
| Supervised. | ResGCN | 70.86±0.45 | 73.21±0.47 | 71.32±1.27 | 73.78±0.43 | 63.53±1.68 | 70.35±1.26 | 84.63±0.95 | 87.45±2.12 | 65.51±1.66 | 71.16±0.74 | 10.3 |
| CL. | ContextPred | 71.24±0.32 | 73.11±0.15 | 72.51±0.61 | 74.82±0.24 | 64.82±2.33 | 70.29±0.52 | 85.74±1.15 | 89.69±2.43 | 68.52±1.78 | 72.30±1.37 | 7.4 |
| | InfoGraph | 72.97±0.85 | 74.21±0.27 | 72.86±0.72 | 75.94±0.41 | 65.27±2.64 | 71.69±0.47 | 86.76±1.34 | 90.33±1.03 | 69.75±1.24 | 74.91±1.45 | 4.5 |
| | GraphCL | 73.39±0.18 | 75.86±0.42 | 73.25±1.24 | 75.84±1.31 | 67.91±1.86 | **73.75±0.34** | 86.63±2.11 | 89.80±1.34 | 69.92±1.87 | 75.26±2.51 | 3.7 |
| | JOAO | 75.86±0.87 | 76.14±0.97 | **73.82±1.48** | 75.52±0.71 | **68.71±0.94** | 72.98±0.72 | 86.81±1.72 | 90.67±1.80 | 70.94±0.95 | 75.66±1.22 | 2.3 |
| Self-training. (DA) | Π-Model | 70.16±0.66 | 71.82±0.97 | 71.56±1.84 | 72.86±1.23 | 62.74±2.72 | 69.74±1.21 | 86.01±1.19 | 88.68±2.72 | 64.77±2.76 | 72.26±1.54 | 10.7 |
| | Pseudo-Label | 70.87±0.49 | 72.31±0.25 | 70.12±1.36 | 71.52±0.46 | 63.22±1.26 | 69.65±2.04 | 86.35±1.39 | 88.91±2.76 | 65.22±1.23 | 72.18±1.64 | 10.6 |
| | Mean Teacher | 71.01±0.29 | 73.51±0.32 | 71.28±1.87 | 73.54±1.24 | 64.39±2.10 | 70.83±1.86 | 86.69±1.40 | 89.60±1.39 | 66.32±2.13 | 71.94±1.41 | 8.3 |
| | DAST | 72.12±1.40 | 75.27±0.34 | 72.45±1.26 | 73.92±1.49 | 64.54±1.36 | 71.01±1.30 | 86.62±1.27 | 89.72±1.82 | 66.92±2.64 | 72.02±1.67 | 6.7 |
| Self-training. (DS) | MPL | 71.27±0.76 | 73.28±0.58 | 71.62±1.55 | 73.45±1.61 | 63.81±2.27 | 70.62±1.78 | 86.74±2.08 | 89.44±2.52 | 66.17±2.36 | 71.64±1.38 | 8.4 |
| | InfoGraph-Semi | 74.23±0.20 | 75.77±1.02 | 73.46±1.85 | 75.11±1.46 | 66.21±2.08 | 73.27±0.75 | 87.35±1.42 | 90.99±1.80 | 70.36±2.46 | 74.82±2.03 | 3.2 |
| | DAST+ | **77.62±0.65** | **78.76±0.71** | 73.58±1.34 | **76.19±1.92** | 67.72±1.18 | 72.98±0.82 | **89.99±0.69** | **92.22±1.21** | **71.50±1.20** | **75.80±1.53** | **1.6** |

Table 7: Overall prediction performance on CIFAR-10, CIFAR-100, STL-10 image datasets with different amounts of labeled data. We report the mean test accuracy with five different seeds. The bold denotes the highest mean score.

| Type | Method | CIFAR-10 | | CIFAR-100 | | STL-10 |
|---|---|---|---|---|---|---|
| | | 1000 | 4000 | 2500 | 4000 | 1000 |
| Supervised. | WRN | 78.12±0.54 | 81.05±0.28 | 39.83±0.39 | 64.83±0.31 | 71.34±0.73 |
| CL. | MOCO | 78.14±0.86 | 82.07±0.67 | 52.82±0.54 | 69.41±0.45 | 81.76±0.71 |
| | SimCLR | 79.25±0.78 | 81.95±0.39 | 54.61±0.52 | 68.72±0.33 | 80.29±0.87 |
| Self-training. (DA) | Π-Model | 72.19±1.07 | 83.03±0.44 | 38.29±0.41 | 62.94±0.25 | 72.88±0.69 |
| | Pseudo-Label | 75.78±0.63 | 82.57±0.31 | 36.54±0.36 | 62.71±0.27 | 70.25±0.62 |
| | Mean Teacher | 80.79±0.42 | 84.17±0.29 | 40.21±0.53 | 66.09±0.24 | 74.56±1.27 |
| | DAST | 81.32±0.35 | 84.25±0.20 | 41.09±0.24 | 66.35±0.25 | 74.91±0.76 |
| Self-training. (DS) | ReMixMatch | 92.31±0.13 | 93.28±0.16 | 67.85±0.26 | 73.04±0.21 | 93.17±0.45 |
| | FixMatch | 92.92±0.24 | 94.17±0.15 | 67.42±0.21 | 72.97±0.17 | 93.25±0.65 |
| | MPL | 93.29±0.29 | 94.21±0.34 | 68.55±0.38 | 72.90±0.26 | 93.21±0.63 |
| | SimMatch | 93.42±0.25 | 94.51±0.32 | 68.95±0.42 | 72.82±0.31 | 93.37±0.72 |
| | DAST+ | **93.74±0.35** | **94.81±0.27** | **69.95±0.48** | **73.15±0.32** | **93.81±0.79** |

are used as the strong data augmentations. We generate pseudo labels using the model's predictions on weakly-augmented unlabeled images and train the classification head to predict the pseudo labels for strongly-augmented images for $\mathcal{L}_{cls}$. Following (Chen et al., 2020a; He et al., 2020), we simply treat the weakly-augmented and strongly-augmented view of the same image as positive contrastive pairs while the augmented data from the different images as the negative pairs for contrastive loss $\mathcal{L}_{cont}$. The reason is that it has been proved the data augmentations in image domain would not change the semantics of the images. Finally, we train DAST+ with the sum loss $\mathcal{L}$ in Eq. (3) using the same experimental settings in DAST.

**Results in image domain.** Table 7 provides the complete comparison results for image classification. We have the following important observations. (1) DAST consistently outperforms other domain-agnostic self-training approaches by achieving 0.42 higher average accuracy than the second best method. This also demonstrates the generality of DAST. (2) DAST performs worse than both domain-specific self-training methods and contrastive learning methods which employ useful data augmentations in image domain. This is reasonable as an effective data augmentation is beneficial for learning better data representations as well as obtaining accurate classifier. (3) DAST+ achieves the highest accuracy across all the datasets. This suggests that existing domain-specific data augmentations can be used as add-ons for DAST to further improve the model performance. Moreover, it also shows that learning better data representations via contrastive learning also improves the performance of domain-specific self-training approaches.

## E    ABLATION STUDY

In this section, we provide the ablation study results on tabular datasets with 10% label ratio to further demonstrate the effectiveness of DAST in under-explored domains, while the results on 5% label ratio have similar trends. Specifically, we first conduct the ablation study to verify the importance of

Table 8: Ablation study on seven tabular datasets with 10% label ratio.

| Ablation \ Dataset | Adult | KDD-cup | CTR-small | Diabetes | Mfeat | Covertype | Solar-flare | Avg. |
|---|---|---|---|---|---|---|---|---|
| DAST- | 88.58 | 69.12 | 61.96 | 82.98 | 91.16 | 89.68 | 68.62 | 78.88 |
| DAST w/o $\mathcal{L}_{cont}$ | 89.26 | 70.61 | 62.87 | 83.05 | 92.14 | 89.96 | 68.93 | 79.55 |
| DAST w/o propagation | 89.51 | 71.04 | 63.38 | 84.07 | 93.02 | 89.92 | 69.27 | 80.03 |
| DAST | 89.95 | 71.75 | 64.26 | 84.93 | 93.23 | 90.47 | 69.90 | 80.62 |

our proposed contrastive learning module and two-way pseudo label generation strategy. Second, we present the accuracy of the selected pseudo labels during self-training process. Then we tune the update frequency $T$ in Section 3.3 to show the effect of the update frequency of pseudo labels on the final results. Finally, we show the effects of hyperparameters of DAST. All the experimental settings are the same as being described in Appendix C.

**Effect of contrastive learning module and pseudo label generation strategy.** We evaluate the effects of the contrastive learning module and our two-way pseudo label generation strategy. Specifically, we consider (1) **DAST w/o propagation** which only utilizes the pseudo labels $\hat{y}^c$ generated from the classification head to train the whole model (replace $\hat{y}^p$ with $\hat{y}^c$), (2) **DAST w/o $\mathcal{L}_{cont}$** which removes the contrastive learning and uses the data embeddings generated from the encoder $\phi(x)$ to produce $\hat{y}^p$. (3) **DAST-** which only trains the classification head with the pseudo labels $\hat{y}^c$.

As shown in Table 8, both DAST w/o $\mathcal{L}_{cont}$ and DAST w/o propagation consistently outperform DAST-, which indicates the effectiveness of the contrastive learning module and two-way pseudo label generation strategy. DAST w/o propagation achieves the second highest accuracy. This is reasonable since we still use the supervised contrastive loss to learn aligned data representations, leading to better model performance. Furthermore, DAST achieves the highest test accuracy across all the datasets. This demonstrates that the contrastive learning and self-training in DAST are manually beneficial.

Table 9: Accuracy of the selected pseudo labels on tabular datasets.

| Pseudo labels \ Dataset | Adult | KDD-cup | CTR-small | Diabetes | Mfeat | Covertype | Solar-flare |
|---|---|---|---|---|---|---|---|
| Pseudo-Label | 93.68 | 95.02 | 94.72 | 99.15 | 99.82 | 93.81 | 99.79 |
| $\hat{y}^c$ | 94.82 | 95.48 | 94.94 | 99.42 | 99.52 | 94.81 | 100 |
| $\hat{y}^p$ | 94.26 | 95.61 | 96.81 | 100 | 99.95 | 93.92 | 99.87 |

**Accuracy of the selected pseudo label.** To show the effectiveness of our proposed label propagation strategy in DAST, we evaluate the precision of the selected pseudo labels during model training on seven tabular datasets. As shown in Table 9, we can find that both the selected $\hat{y}^c$ and $\hat{y}^p$ achieve high precision, which means most of the pseudo labels used for model training are the ground-truth labels. This demonstrates that our proposed label propagation has a strong ability to select the reliable contrastive pairs for learning better data representations and high-quality pseudo labels for training the classification head. Besides, We have found that the accuracy of $\hat{y}^p$ consistently outperforms the accuracy of the selected pseudo labels in Pseudo-Label, which implies that only using one classification head would cause direct error accumulation, resulting in inaccurate pseudo labels. The pseudo labels selected in DAST are more reliable since they are identified with high confidence by both the classification head and the projection head. Our two-head structure can avoid direct error accumulation, thus improving the accuracy of the selected pseudo labels in both heads.

Table 10: Performance of DAST with varying $T$ on three large tabular datasets.

| Method | Adult | | KDD-cup | | Covertype | |
|---|---|---|---|---|---|---|
| | Time Cost | Accuracy | Time Cost | Accuracy | Time Cost | Accuracy |
| DAST ($T = 1$) | 1941s | 90.01 | 8433s | 71.82 | 6339s | 90.53 |
| DAST ($T = 5$) | 640s | 89.95 | 5716s | 71.75 | 2275s | 90.47 |
| DAST ($T = 10$) | 549s | 89.72 | 5377s | 68.95 | 1771s | 90.04 |

**Effect of the pseudo label update frequency.** We select the three largest tabular datasets to show the impact of the update frequency of pseudo labels on time cost and prediction accuracy, while the other small datasets have similar trends. As shown in Table 10, we find that decreasing $T$ improves the predictive performance of DAST. This is because a smaller $T$ allows the model to update the pseudo labels more frequently, which can avoid error accumulation of inaccurate pseudo labels. Besides, we observe that decreasing $T$ would greatly increase the time cost, and the improvement of predictive performance decreases as $T$ becomes smaller. Specifically, when $T$ decreases from 5 to 1, the average improvement of accuracy is merely 0.06 but the additional time cost is 2694s. Hence, we suggest to choose reasonably small values for $T$ to balance the computational efficiency and predictive performance based on the application requirements.

Table 11: Effect of hyperparameter $\beta$ on tabular datasets.

| Ablation \ Dataset | Adult | KDD-cup | CTR-small | Diabetes | Mfeat | Covertype | Solar-flare |
|---|---|---|---|---|---|---|---|
| $\beta = 0.1$ | 89.27 | 71.26 | 63.74 | 83.07 | 92.48 | 89.93 | 69.57 |
| $\beta = 0.3$ | 89.52 | 71.69 | 64.05 | **84.93** | **93.23** | 90.31 | **69.90** |
| $\beta = 0.5$ | **89.95** | **71.75** | **64.26** | 84.70 | 93.09 | **90.47** | 69.48 |
| $\beta = 1.0$ | 89.44 | 71.58 | 64.09 | 84.41 | 93.07 | 90.25 | 69.31 |

Table 12: Effect of hyperparameter $T_{warm}$ on tabular datasets.

| Ablation \ Dataset | Adult | KDD-cup | CTR-small | Diabetes | Mfeat | Covertype | Solar-flare |
|---|---|---|---|---|---|---|---|
| $T_{warm} = 0$ | 89.52 | 71.55 | 64.03 | 84.75 | 93.07 | 90.28 | 69.52 |
| $T_{warm} = 20$ | 89.84 | 71.62 | **64.26** | **84.93** | **93.23** | 90.34 | **69.90** |
| $T_{warm} = 40$ | **89.95** | **71.75** | 64.12 | 84.79 | 93.13 | **90.47** | 69.58 |

**Effect of the hyperparameter $\beta$ and $T_{warm}$.** We conduct the experiments to show the effectiveness of $\beta$ and provide the results in Table 11. Recall that $\beta$ represents the trade-off between using predictions from the classification head and those from the projection head to select the pseudo labels for the classification head. Particularly, $\beta = 1(0)$ means only using the pseudo labels generated from the projection head (classification head) to select the pseudo labels for the classification head. We observe that $\beta = 0.3$ or $\beta = 0.5$ achieves the best model performance. This is because the selected pseudo labels are more reliable when the pseudo labels generated by the classification head and the projection head are consistent and are both with high confidence. $\beta = 0.1$ performs the worst since mainly using the pseudo labels generated from the classification head to update model parameters would cause direct error accumulation in the classification head. Besides, $\beta = 1$ performs worse than $\beta = 0.5$. This is because the learned data representations in the early stage of model training are relatively not aligned. Thus, only generating the pseudo labels based on such data representations are less accurate, resulting in the degeneration of the model performance.

$T_{warm}$ denotes the starting epoch of unsupervised learning and we show the effect of different values of $T_{warm}$ in Table 12. We observe that $T_{warm} = 0$ performs the worst. This is because the predictions of both the classification head and the projection head are inaccurate at the beginning of training, resulting in inaccurate pseudo labels that hurt model performance. $T_{warm} = 20$ performs better than $T_{warm} = 40$. The reason may lie in the fact that a long warm-up epoch for some datasets would cause overfitting.

It is worth mentioning that while tuning all the hyperparameters may achieve higher performance, it increases engineering efforts significantly. To avoid expensive engineering efforts, we propose to fix the other hyperparametes (except $\beta$ and $T_{warm}$) so that our proposed framework can be applied to a new domain more efficiently.

## F    COMPARISON WITH TRADITIONAL MACHINE LEARNING MODELS

We further compare DAST with the traditional machine learning models which are competitive on tabular datasets (Gorishniy et al., 2021). In particular, we choose Xgboost (Chen & Guestrin, 2016), CatBoost (Prokhorenkova et al., 2018), and Logistic Regression as the baselines. We conduct the experiments on seven tabular datasets with 10% label ratio. As shown in Table 13, DAST outperforms

three traditional machine learning models. This implies that machine learning models may fail in semi-supervised learning with a small number of labeled samples, while they are competitive with sufficient labeled training samples.

Table 13: The comparison results with supervised learning methods on tabular datasets.

| Method \Dataset | Adult | KDD-cup | CTR-small | Diabetes | Mfeat | Covertype | Solar-flare | Avg. |
|---|---|---|---|---|---|---|---|---|
| Xgboost | 88.57 | 64.56 | 63.55 | 84.98 | 92.56 | 89.99 | 69.61 | 79.12 |
| CatBoost | 87.49 | 65.61 | 64.14 | 85.15 | 92.86 | 89.40 | 70.43 | 79.29 |
| Logistic Regression | 87.95 | 64.91 | 62.78 | 84.52 | 92.02 | 89.08 | 68.91 | 78.59 |
| DAST | 89.95 | 71.75 | 64.26 | 84.93 | 93.23 | 90.47 | 69.90 | 80.62 |

## G    COMPLEXITY ANALYSIS

As for the additional computational cost of two-way label generation strategy, constructing a graph of all neighbors requires computing $A^\top A$, where $A \in R^{n*d}$ and $n$ represents the amount of data and $d$ denotes the representation dimension. The complexity of matrix multiplication is $O(n^2)$ since $d << n$ (the maximum $d$ used in DAST is 32). Following (Chandra & Kokkinos, 2016; Iscen et al., 2017), by employing effective parallel computing methods, such computational cost is acceptable even for large-scale datasets with more than $10^6$ samples. Besides, as mentioned in Section 3.2, we follow Iscen et al. (2017) and utilize conjugate gradient (CG) method for computing the diffusion matrix, which has an affordable complexity of $O(n^2)$.

As for the memory cost of the affinity matrix $G$, we only need to store the similarity values of k-nearest neighbors for each data point, resulting in a total storage cost of $k * N$ . In our experiments, we set the maximum of $k$ to be 30, which will not incur large memory overhead.

## H    ADDITIONAL DISCUSSIONS

**Limitations.** In the main paper, we have discussed a novel self-training framework DAST which is effective for domains without effective data augmentations. Based on the proposed contrastive learning module and two-way pseudo label generation strategy, DAST liberates the reliance on data augmentations and generates reliable pseudo labels based on highly distinguishable data representation. However, the additional contrastive learning module and two-way pseudo label generation strategy would cause additional computational cost, which influences the training efficiency. So far, as discussed in Appendix E, we utilize the hyperparameter $T$ to control the trade-off between the accuracy and training time cost. We leave the further improvement of training efficiency as the future work.

**Difference with domain adaption methods.** Domain adaption aims to transfer the knowledge learned from one or more related domains to an target domain (Long et al., 2015; Farahani et al., 2021; Ding et al., 2022). It is motivated by the challenge where the test and training domains have different data distributions. However, our method focus on the semi-supervised learning in a domain without effective data augmentations. The domain adaptation which lies in transfer learning is beyond the scope of this work.

