# OpenReview forum: "Domain-Agnostic Self-Training for Semi-Supervised Learning"
_ICLR.cc/2024/Conference — ICLR 2024 Conference Withdrawn Submission_

### Official Review · Reviewer_RRjm · 2023-10-30

**Soundness:** 3 good
**Presentation:** 3 good
**Contribution:** 2 fair
**Rating:** 5
**Confidence:** 5

**Summary:**

This paper introduces a generic self-training framework that is applicable to domains without effective data augmentations. DAST decouples the generation and utilization of pseudo labels by incorporating the classification module with a contrastive learning module. Numerous experiments have proved its effectiveness.

**Strengths:**

This paper makes a combination of self-supervised and semi-supervised learning for pseudo label choosing and has good experimental performance.

**Weaknesses:**

This paper does not discuss out-of-distribution samples and long-tailed distributions, and I would like to see experimental validation as to whether the framework proposed in this paper can still outperform existing methods in these two cases.

Is the nearest neighbor operation time-consuming and would like to see further clarification.

This paper makes an analysis of alignment in contrastive learning, but there is no discussion of uniformity. I would like to see a discussion on whether the introduction of comparative learning will result in dimensional collapse on top of the original models.

**Questions:**

The author seems to be elucidating that data augmentation is not a good semi-supervised learning tool, I'm curious what the problem with data augmentation is, as a means of consistency regularization, you can learn a better prototype with data augmentation, why not? I would like to see further explanation as to why methods that rely on data augmentation are called DOMAIN-AGNOSTIC and methods that do not rely on data augmentation are called DOMAIN-AGNOSTIC.

Is formula 7 sensitive to alpha?

What are your advantages over a DOUBLE THRESHOLD approach like UPS, I'd love to see further elaboration.

Please add a set of experiments, using a self-supervised pre-trained backbone as the initial model, and then select pseudo label, I think this set of experiments is very necessary, and this set of experiments can show how the introduction of self-supervision has improved your methods, since many of the other methods you are comparing do not take self-supervision into account.

**Details Of Ethics Concerns:**

None.

---

### Official Review · Reviewer_e71P · 2023-10-30

**Soundness:** 2 fair
**Presentation:** 3 good
**Contribution:** 2 fair
**Rating:** 3
**Confidence:** 5

**Summary:**

This paper introduces the DAST model, aiming to address domain-agnostic semi-supervised learning. Its core idea is to decouple the generation and utilization of pseudo-labels. The model consists of a feature extractor, classification head, and projection head. A contrastive learning module is designed, along with a two-way pseudo-label generation strategy. In addition, this paper also provides theoretical justification for the effectiveness of the model from the perspective of the Expectation-Maximization (EM) algorithm.

**Strengths:**

1. The paper has a well-structured and logical organization, and the experimental section is comprehensive in terms of its content.

2. The figures in the paper provide clear and concise explanations, allowing readers to quickly grasp the specific structure of the model and understand the differences between various self-training methods.

**Weaknesses:**

However, the paper has the following shortcomings:

1. Lack of innovation: The model design bears a striking resemblance to the paper "PiCO: Contrastive Label Disambiguation for Partial Label Learning." Both papers share similar ideas such as decoupling, contrastive learning modules, and label disambiguation, with theoretical analyses approached from an EM perspective.

2. Insufficient coverage of the domain-agnostic problem in the related work section.

3. In the ablation experiments, the observed effects are minimal, with improvements of less than one percent, indicating weak results.

4. The baselines used in the experiment section for DA self-training belong to early works which are the most compatible with the setting of this paper, making them less persuasive.

**Questions:**

See Weaknesses.

---

### Official Review · Reviewer_yyif · 2023-10-31

**Soundness:** 2 fair
**Presentation:** 2 fair
**Contribution:** 3 good
**Rating:** 3
**Confidence:** 4

**Summary:**

Self-training, or pseudo-labeling, is a powerful and widespread semi-supervised technique nowadays. However, recent advanced pseudo-labeling algorithms heavily rely on the data augmentation available for a domain, e.g. images, speech and text, to achieve state-of-the-art results. In this paper authors propose variant of pseudo-labeling, dubbed DAST, which is agnostic to a domain and does not rely on the augmentation. Instead, DAST performs joint cross-entropy and contrastive loss training with pseudo-labels generated by the other head of the network for each loss, emulating EM algorithms and forcing minimization of intraclass distance in the representation clustering. Authors show this EM intuition and provide theoretical justification behind it. With empirical results authors compare DAST with prior methods on tabular data where augmentation is not readily available as well as on graph and image data where augmentation can be incorporated into DAST too. Results show that DAST outperforms domain-agnostic, domain-specific and contrastive methods.

**Strengths:**

- Pushing for domain-agnostic pseudo-labeling without reliance on data augmentation as many domains do not have it
- Idea on having parallel representation and classification embeddings which gives each other pseudo-labels: we try to align both classification and representation tasks from the pseudo-labels perspective.
- Empirical analysis on tabular, graph and image data to confirm hypothesis.
- Interpretation as EM-algorithm for the proposed method.
- Details on the hyper parameters search in Appendix.

**Weaknesses:**

- in the main text there is no discussion between methods of continuous pseudo-labeling when one model is trained with time to time regenerated pseudo-labels and teacher-student approach. The latter is the simplest domain agnostic method which works well in practice in many domains, including tabular data. This distinction between methods is needed as they have different training dynamic and different problems (e.g. it was shown recently that FixMatch can diverge for transformer-based models in vision  Cai, Z., Ravichandran, A., Favaro, P., Wang, M., Modolo, D., Bhotika, R., ... & Soatto, S. (2022). Semi-supervised vision transformers at scale. Advances in Neural Information Processing Systems, 35, 25697-25710.)
- There is no comparison with plain teacher-student baseline to understand if the issue of all continuous methods is exactly data augmentation and /or continuous nature of training.
- There is no detailed disambiguation in the proposed method between cross-entropy loss with pseudo-labels and contrastive loss, e.g. what is the quality of supervised training when contrastive loss is not used or is used, what about other methods when contrastive loss is added? Does improvement come from additional contrastive loss or from new scheme of pseudo-labels cross usage? (Table 8 only partially answers these questions, and then you need to compare with self-supervised + self-training methods as a baseline).
- In the projection head, pseudo-labels updating is proposed to do every T steps. Authors did not study how this affects training stability. Could it be that this step is "must" for the training, as otherwise both pseudo-labels in E and M steps are changing too fast causing divergence as in prior works, e.g. Cai, Z., et al. Semi-supervised vision transformers at scale. Advances in Neural Information Processing Systems, 35, 25697-25710 or Likhomanenko, T, et al. "slimipl: Language-model-free iterative pseudo-labeling." Interspeech 2021?
- Errors in the proof of Theorem 1:
  - second part of the last part of eq (14) is wrong, there should be log sum exp operation
  - in eq (16) first row: there is missing |S_j|
  - why data embeddings are on the hypersphere? I do not see any conditions on that or normalisation in the network.

**Questions:**

Typos:
- Abstract: "images.)" -> "images" and "can also combine" -> "can also be combined"


Missing details / questions / suggestions:
- What is fully supervised model upper bound quality?
- In the text I feel overstatement on the quality of pseudo-labels, as later authors say that they use filtering. Initial statement should be smoother then.
- In the text I feel overstatement of error propagation, as now it is indirect and more complicated dependency could be on error propagation if any part of the algorithm is broken (e.g. if contrastive part is broken).
- Label propagation is not clearly articulated in the text. Formulas are good, but some description what exactly is behind these formulas will be helpful, especially due to usage of classification head pseudo-labels in this process.
- Table 1: why contrastive is worse than supervised training with labeled data only?
- Does DAST work better in low data regime as shown in the paper? What happens with larger data regime as maybe contrastive is not so helpful there?
- SimCLR results in Table 3 are very poor compared to the original paper, why is it?
- In Table 13: are all boosting methods trained in fully supervised regime?
- Additional citation to check https://openreview.net/forum?id=TAVBJ4aHsWt